# Paradoxical Roles of Oxidative Stress Response in the Digestive System before and after Carcinogenesis

**DOI:** 10.3390/cancers11020213

**Published:** 2019-02-13

**Authors:** Akinobu Takaki, Seiji Kawano, Daisuke Uchida, Masahiro Takahara, Sakiko Hiraoka, Hiroyuki Okada

**Affiliations:** Department of Gastroenterology and Hepatology, Okayama University Graduate School of Medicine, Dentistry and Pharmaceutical Sciences, Okayama 700-8558, Japan; skawano@mpd.biglobe.ne.jp (S.K.); d.uchida0309@gmail.com (D.U.); m_takahara009@yahoo.co.jp (M.T.); sakikoh86@yahoo.co.jp (S.H.); hiro@md.okayama-u.ac.jp (H.O.)

**Keywords:** oxidative stress, antioxidant, cancer, reactive oxygen species

## Abstract

Oxidative stress is recognized as a cancer-initiating stress response in the digestive system. It is produced through mitochondrial respiration and induces DNA damage, resulting in cancer cell transformation. However, recent findings indicate that oxidative stress is also a necessary anticancer response for destroying cancer cells. The oxidative stress response has also been reported to be an important step in increasing the anticancer response of newly developed molecular targeted agents. Oxidative stress might therefore be a cancer-initiating response that should be downregulated in the precancerous stage in patients at risk of cancer but an anticancer cell response that should not be downregulated in the postcancerous stage when cancer cells are still present. Many commercial antioxidant agents are marketed as “cancer-eliminating agents” or as products to improve one’s health, so cancer patients often take these antioxidant agents. However, care should be taken to avoid harming the anticancerous oxidative stress response. In this review, we will highlight the paradoxical effects of oxidative stress and antioxidant agents in the digestive system before and after carcinogenesis.

## 1. Introduction

Chronic inflammatory disease is associated with a risk of carcinogenesis in many digestive organs. Examples include chronic atrophic gastritis for gastric cancer, inflammatory bowel disease (IBD) for colitic cancer, chronic hepatitis for hepatocellular carcinoma (HCC) and cholangiocellular carcinoma, and chronic pancreatitis for pancreatic cancer. The rates of infection-related diseases, such as chronic gastritis (*Helicobacter pylori*; *H. pylori*) and chronic hepatitis (hepatitis B and C viruses), are decreasing with the advent of antimicrobial treatments, while the rates of other etiology-related diseases are increasing. Representative non-infection-related chronic inflammatory digestive diseases include IBD, chronic pancreatitis and nonalcoholic fatty liver disease (NAFLD) [1,2].

Oxidative stress is a cellular stress associated with inflammation or environmental stressors. The most studied source of oxidative stress is reactive oxygen species (ROS). Inflammation or environmental stressors—such as smoking or lipid overload—can induce ROS via neutrophils and macrophage activation. These stressors can also induce the resident cells to produce ROS through microsomes, peroxisomes, and activation or damage of the mitochondrial energy metabolism pathway [3]. The mitochondrial respiratory pathway and the nicotinamide adenine dinucleotide phosphate (NADPH) oxidase (Nox) enzyme pathway are the two major producers of endogenous ROS [4]. Approximately 90% of ROS are produced during the mitochondrial respiration process; the Nox-related transport of electrons across the membranes accounts for the rest [5].

Physiological levels of ROS are a natural response in animals and plants to restore a toxic microbiome and are required in the plasma membrane repair response [6].

However, chronic exposure to high levels of ROS induces various events that are related to not only the progression of chronic inflammatory disease but also to carcinogenesis, including DNA damage, tissue remodeling and gene expression changes [7,8]. ROS damage DNA resulting in the production of 8-oxo-2′-deoxyguanosine (8-oxo-dG) which is found in the early phases of carcinogenesis [9]. Lipids are another target of ROS. The peroxidation of lipids results in the production of 4-hydroxynonenal (4-HNE) or malondialdehyde (MDA) induces angiogenesis or carcinogenic Wnt/β-catenin and PI3K/Akt signaling activation [10]. Tumor suppressor gene PTEN is inactivated by H_2_O_2_ (an ROS) in a time- and concentration-dependent manner [11]. In addition to these cellular carcinogenic activities, many cytotoxic anticancer agents induce the oxidative stress to damage cancer cells [12]. These contradictory findings prompt confusion concerning the appropriate regulation of oxidative stress before and after carcinogenesis.

The function of antioxidants is also involved in carcinogenesis. Heme oxygenase-1 (HO-1), an antioxidant molecule, is an enzyme that catalyzes the oxidative degradation of cellular heme into free iron, carbon monoxide, and biliverdin, which is then converted into nontoxic bilirubin. Targeting HO-1 to reduce the antioxidant function of cancer cells is a recent approach in cancer treatment [13].

## 2. Perspective of this Review

Oxidative stress is a necessary response in human health; however, in the setting of chronic inflammatory disease, it becomes toxic and is involved in carcinogenesis. However, oxidative stress is favorable for cancer patients as it is also toxic to cancer cells. These contradictions are very difficult to understand. At present, there is no way to stage patients according to their oxidative-antioxidant status and there is no established approach to the management of oxidative stress-related conditions. In present review, we would like to unveil the pros and cons of oxidative stress and the administration of antioxidant agents before and after carcinogenesis.

## 3. Current Understanding of Oxidative Stress in the Digestive System

Chronic inflammatory disease is associated with a risk of carcinogenesis in many digestive organs. Examples include chronic gastritis, inflammatory bowel diseases, and chronic liver diseases. The overview of the oxidative stress in the digestive system is summarized in this section.

### 3.1. Oxidative Stress in Chronic Digestive Diseases

The progression of chronic infection-related diseases has been shown to be correlated with oxidative stress in humans, mice and in vitro [14]. *H. pylori*, a Gram-negative microaerophilic bacterium, has been shown to induce oxidative stress in the infected gastric mucosa, thereby contributing to the development of chronic atrophic gastritis and gastric cancer. The pathogenic mechanisms of HBV- and HCV-related chronic liver disease (CLD) and hepatocarcinogenesis include viral protein-related immune function interference, tumor initiation or suppression interference, and CLD-related environmental changes. Oxidative stress is involved in this process via the direct effects of viral proteins or secondary to chronic inflammation [15].

Gastric cancers can develop after *H. pylori* eradication, and HCC can develop after hepatitis virus eradication. Gastric mucosal atrophy and liver fibrosis can continue even after microbial eradication, contributing to carcinogenesis. Environmental stressors, such as obesity-related oxidative stress, have been acknowledged to affect the clinical course. Non-microbial chronic digestive inflammatory diseases may initiate carcinogenesis even without concomitant microbial stimulation.

Obesity-related fat deposition, hyperglycemia, and hyperlipidemia can initiate and exacerbate carcinogenesis. In recent years, increasing numbers of people have been adopting high-fat, high-lipid, and high-fructose diets with artificial sweeteners from infancy. Such a condition has never been experienced before; thus, the eventual outcome of such a lifestyle is unclear at present. Experimental data might help clarify the future of modern populations living such lifestyles. The deposition of lipids in the stomach and liver induces oxidative stress and chronic inflammation [16,17].

### 3.2. Problems in the Management of Oxidative Stress

There are many unresolved issues regarding the evaluation and management of oxidative stress in the clinical setting. Studies should be performed to establish appropriate protocols for the management of oxidative stress in patients with chronic digestive disease or cancer.

#### 3.2.1. How to Monitor Oxidative Stress

Monitoring oxidative stress in in vivo models and patients is difficult due to the complex nature of the oxidative and antioxidative balance and the very short half-life of ROS [18]. There are many assays to measure oxidative stress and the antioxidant functional reservoir indirectly. One of the most studied oxidative stress markers is 8-OHdG, the level of which reflects the amount of oxidized DNA. There are also other markers, including MDA, 4-HNE, oxidized low-density-lipoproteins, and reactive oxygen metabolites (ROM). Antioxidant markers are also available, including enzymatic markers (catalase, GPx, and superoxide dismutase) and nonenzymatic markers (vitamin E, A, C, and uric acid). Currently there is no consensus on the optimal methods for assessing individual oxidative stress-related conditions [19].

#### 3.2.2. The Control of Oxidative Stress

The control of oxidative stress in patients depends on the patient’s condition, including sex, age, baseline chronic disease, and cancer stage. Antioxidant strategies are divided into two groups: the first strategy is to modulate or stabilize ROS via the activation of antioxidative stress-related pathways such as the Nrf2 pathway and the second strategy is to remove reactive intermediates [20]. Given that the pharmacokinetics of antioxidants are unfavorable for reaching the mitochondria, the modulating of ROS-related pathways maybe beneficial. Strategies to inhibit the formation of ROS are more promising than ROS scavenging.

Several prospective studies investigating the effects of antioxidants in the prevention of cancer or mortality have shown conflicting results. The Alpha-Tocopherol, Beta-Carotene Cancer Prevention Study (ATBC) showed that alpha-tocopherol decreased the incidence of prostate cancer, whereas beta-carotene increased the risk of lung cancer and total mortality [21]. However, one RCT—the Selenium and Vitamin E Cancer Prevention Trial (SELECT)—which assessed the risk of prostate cancer with vitamin E administration, found a 17% increase in the incidence of prostate cancer [22]. The Beta-Carotene and Retinol Efficacy Trial (CARET) showed an increased risk of lung cancer [23]. An epidemiologic study in China reported dietary vitamin E intake and vitamin E supplement use was associated with reduced risks of liver cancer, while vitamin C and multivitamin intake increased the risk [24]. Antioxidant supplementation might be useful for selected populations. However, there are limited data on the effect of antioxidant supplementation on the risks of gastrointestinal cancer.

## 4. Oxidative Stress in the Upper Gastrointestinal Tract

The role of oxidative stress in gastric cancer has been considered from two aspects. *H. pylori* leads to chronic inflammation due to failed eradication. Several components of this bacterium, including cytotoxin-associated gene A (cagA)-encoded CagA protein, play direct roles in inducing chronic inflammation and carcinogenesis by causing oxidative stress [25]. This is a major contributor to DNA damage, apoptosis, and neoplastic transformation. However, the production of ROS by cancer cells also plays an important role in their eradication (Figure 1).

Pregastric cancer state: The ROS level in *H. pylori*-positive patients with chronic gastritis is higher than that in *H. pylori*-negative patients with chronic gastritis, likely due to the presence of CagA protein. CagA protein is localized to the mitochondria, where it subsequently suppresses tumor suppressor sirtuins (SIRT) 3 resulting in increased ROS and the hypoxia-related transcription factor hypoxia-inducible factor 1α (HIF-1α). A carcinogenesis-related protein aquaporin 3 (AQP3) promoter contains a putative hypoxia response element (HRE) that could be controlled by hypoxia and HIF-1α. The ROS-HIF-1α-AQP3-ROS loop may thus be an important factor leading to the development of gastric cancer. CagA protein also has the function to activate Wnt/β-catenin signal resulting in cell proliferation and carcinogenesis.

Cancer state: In the cancer state, oxidative stress is included in anticancer responses. A pharmacologic dose of ascorbic acid has been shown to be associated with the increased efficacy of chemotherapeutics and radiation treatment. Cancer stem cells are considered to be resistant to therapy because they have an enhanced protection against ROS. The CD44 variant nine controls the intracellular level of reduced glutathione (GSH), resulting in a higher expression of GSH and defense against ROS. Escaping from ROS is one characteristic of cancer stem cells, suggesting that oxidative stress is an essential reaction for controlling cancer progression and that it should be managed appropriately.

### 4.1. Oxidative Stress in H. pylori-Related Gastric Carcinogenesis: Before Carcinogenesis

*H. pylori* infection was classified as a human gastric carcinogen by the International Agency for Research on Cancer.

The ROS level in *H. pylori*-positive patients with chronic gastritis is higher than that in *H. pylori*-negative patients with chronic gastritis, likely due to pathogen-inherent virulence factors, such as CagA protein, and the type and intensity of oxidative stress induced by inflammation. CagA may be delivered into gastric epithelial cells via the type IV secretion system (TFSS) [26,27]. When the bacterial surface-exposed CagA attaches to the plasma membrane, it is translocalized to the gastric epithelial cells, after which the pathophysiological actions take place [28]. In CagA-positive *H. pylori*-infected gastric epithelial cells, the CagA protein is localized to the mitochondria, where it subsequently suppresses the tumor suppressor sirtuins (SIRT) 3 resulting in increased ROS and the hypoxia-related transcription factor hypoxia-inducible factor 1α (HIF-1α) [29]. The oxidative stress produced by *H. pylori* infection, especially with CagA, not only causes DNA damage but also prevents DNA repair mechanisms from functioning properly, subsequently causing increased apoptosis and cellular proliferation. In addition, CagA upregulates Wnt/β-catenin signaling, resulting in the development of cancer stem cells [30].

The carcinogenesis-related protein aquaporin 3 (AQP3) has recently been shown to be induced via the ROS pathway. Aquaporins (AQPs) are a family of small and integral membrane proteins that transport water or glycerol and hydrogen peroxide across biomembranes [31]. They play an important role in water homeostasis, fat metabolism, urine concentration, exocrine gland secretion, and cancer biological functions. In gastric cancers, the expression of AQP3 has been shown to be positively associated with lymph node metastasis, low histological classification and lymphovascular invasion [32]. The positive function of AQP3 on the oxidative stress production mechanism was proven in retrospective clinical samples as well as in vitro *H. pylori* infection experiments [33]. The AQP3 promoter was shown to contain a putative hypoxia response element (HRE) that could be controlled by hypoxia and HIF-1α, which are representative oxidative stress responses. In addition, *H. pylori* infection induced the production of proinflammatory cytokines, such as IL-6, IL-8, and TNF, depending on the level of AQP3. The ROS-HIF-1α-AQP3-ROS loop may be an important factor leading to the development of gastric cancer.

Given that most patients with *H. pylori* infection have not yet developed gastric cancer, *H. pylori* infection alone is not sufficient to induce the development of gastric cancer. Environmental factors, such as the cigarette smoking, alcohol intake, obesity, and chemical exposure, also increase the ROS level and thereby the risk of developing gastric cancer [34].

Several prospective studies investigated the effects of antioxidant therapy intervention and have shown conflicting data [35,36]. A systematic review and meta-analysis of studies that investigated dietary contents revealed that consumption of fruit and white vegetables was inversely correlated with the gastric cancer risk, suggesting that a vitamin rich diet protects against gastric cancer development [37]. However, a systematic review and meta-analysis of the antioxidant intake concluded that antioxidant supplements could not prevent gastrointestinal cancers, while they seemed to increase overall mortality [36]. A Chinese prospective study revealed that the risk of gastric cancer began to be reduced 1 to 2 years after the initiation of betacarotene, vitamin E, and selenium supplementation [38]. Ten years after finishing the intervention, the effect was maintained with reduced gastric cancer mortality, while the effects were gone after 25 years [39]. It is still difficult to draw conclusions about the effectiveness of antioxidant treatment to prevent the development of gastric cancer.

### 4.2. How to Manage Oxidative Stress in Gastric Cancer: After Gastric Carcinogenesis

After the development of cancer, antioxidant therapy must be administered with care as it might reduce the anticancer response. Analyzing the expression of oxidative stress-related molecules is also confusing such as the mRNA expression of the oxidative stress-related gene NOX2 in gastric cancer was associated with a better prognosis, while the mRNA expression of NOX4 was associated with a poor prognosis [40].

Cancer stem cells are considered resistant to therapy because they have enhanced protection against ROS as epithelial stem cells [41]. Cancer stem cells and epithelial stem cells both exhibit lower ROS than nontumornontumorigenic cells due to the increased expression of the free-radical-scavenging system. These results indicate that an enhanced ROS scavenging system is a characteristic of cancer stem cells. One explanation for this mechanism may involve the function of a cancer stem cell marker, CD44. The adhesion molecule CD44 is known as a cancer stem cell marker [42]. Early gastric cancer patients who exhibit relatively high expression levels of CD44 variant 9 in the gastric epithelium showed early recurrence after endoscopic resection [43]. In mouse gastric tumors, CD44 variant was abundantly and heterogeneously expressed in proliferative cells and slow-cyclin stem-like cells but not in differentiated cells [44]. The CD44 variant interacts with xCT, a—glutamate-cystine transporter—and controls the intracellular level of reduced glutathione (GSH), resulting in a higher expression of GSH and defense against ROS. Escaping from ROS is one characteristic of cancer stem cells, suggesting that oxidative stress is an essential reaction for controlling cancer progression and that it should be managed appropriately.

In cancer stages, approaches to alter oxidative stress conditions are complicated. Standard antioxidants such as ascorbic acid might not be a good approach, while a pharmacologically high dose of ascorbic acid, which could only be achieved via intravenous administration, was reported to have a prooxidant function [45]. A pharmacologic dose of ascorbic acid has been reported to increase toxic H_2_O_2_ and induce the death of several cancer cells [46]. The ascorbic acid radical and H_2_O_2_ may be concentrated in extravascular tumor tissue, and may be erased by red blood cell membrane-reducing proteins and/or by large plasma proteins that are not distributed in the extravascular spaces [47]. This effect was specifically observed against cancer cells and not normal cells or blood. In gastric adenocarcinoma cell experiments, a pharmacologic dose of ascorbic acid which could not be achieved via oral intake, was associated with the increased efficacy of chemotherapeutics and radiation treatment [48]. This approach might be adopted for a future anticancer adjuvant therapy.

## 5. Oxidative Stress in the Colon

Colorectal cancer (CRC) is one of the most common cancers worldwide: the highest incidence is observed in Western countries [49].

Widely accepted sporadic CRC risk factors include aging and lifestyle factors. Colitis-associated CRC risk factor is inflammatory bowel disease (IBD), which includes Crohn’s disease (CD) and ulcerative colitis (UC). According to recent studies, oxidative stress is an important progenitor and is related to genetic and epigenetic changes in the development of CRC (Figure 2) [50,51].

Pre-colon cancer state: ROS is an important initiator for colon cancer. In sporadic-type CRC, life-style related stressors such as smoking, alcohol, and obesity have been reported to induce colon carcinogenesis. In colitic cancer, chronic inflammation due to inflammatory bowel disease (IBD) can induce ROS.

Colon cancer state: The level of oxidative stress has been shown to be elevated in cancer cells in comparison to normal cells. Nontumor tissue has been shown to produce high levels of antioxidant enzymes, possibly to defend against cancer-producing oxidative stress. The expression of antioxidant-related enzymes, such as MnSOD, in stage III and IV CRC tumor samples was shown to be increased, and the expression of several antioxidant enzymes, including catalase and GPx, was also increased in adjacent nontumor tissues. The effect of high-dose ascorbic acid on CRC cell lines is dependent on the expression of a tumor suppressor p53. CRC cells carrying mutant oncogenes of KRAS or BRAF were killed with high dose ascorbic acid. Dehydroascorbate (DHA), the oxidized form of ascorbic acid, was transported into CRC cells via glucose transporter GLUT1, which was overexpressed in KRAS or BRAF mutant CRC. The increased uptake of DHA caused oxidative stress as intracellular DHA was reduced due to ascorbic acid, which depleted glutathione. Vitamin E analogue erased the cancer cell toxic response of ascorbic acid via the recovery of the TCA cycle and ATP production.

### 5.1. Oxidative Stress in Sporadic CRC

Most CRC is sporadic-type, which is believed to be correlated with lifestyle. Thus, exogenous factors, such as food, alcohol consumption, and smoking, are major risk factors. The gastrointestinal (GI) tract is a key source of oxidative stress. As the GI tract is constantly exposed to ingested materials and pathogens, including bacteria, oxidative stress can be induced by an individual’s personal environment [52]. In sporadic-type CRC, the level of oxidative stress has been shown to be elevated in comparison to normal cells, which is attributed to the increased metabolic activity of CRC [51]. The plasma oxidative stress level has also been shown to be increased in sporadic CRC patients [53]. Nontumor tissue has been shown to produce high levels of antioxidant enzymes, possibly to defend against cancer-producing oxidative stress. The expression of antioxidant-related enzymes, such as MnSOD, in stage III and IV CRC tumor samples was shown to be increased, and the expression of several antioxidant enzymes, including catalase and GPx, was also increased in adjacent nontumor tissues [54]. The antioxidant enzyme expression in nontumor tissue adjacent to stage IV tumors was shown to be more strongly activated than that adjacent to stage III tumors, suggesting that the adjacent nontumor tissue helps defend against the strong oxidative stress associated with advanced cancer [54].

Oxidative stress has been shown to affect DNA methylation, resulting in the activation of several carcinogenesis-related genes [55]. ROS-induced oxidized DNA, such as 8-hydroxydeoxyguanosine (8-OHdG) in CpG dinucleotide, can inhibit the methylation of carcinogenesis-related genes [56]. The accumulation of epigenetic changes such as the DNA methylation of tumor suppressor genes in the colon epithelium leads to the development of cancer [57]. ROS are an eraser of such methylation of tumor suppressor genes. CRC is accompanied by the strong expression of antioxidant enzymes and can adapt to a highly oxidized environment. Sustained exposure to excessive oxidative stress and an antioxidant environment can promote oncogenic activity, resulting in cancer development.

Patients with genetically driven CRC, such as in familial adenomatous polyposis (FAP), have been shown to have lower whole blood ROS levels in comparison to patients with sporadic CRC [58]. These results suggest that oxidative stress may play a strong role in sporadic CRC but not in FAP.

### 5.2. Oxidative Stress in Colitic Carcinogenesis

Chronic inflammation is a major risk factor for carcinogenesis, and roughly 15–20% of cases of carcinogenesis are reported to occur as a result of chronic inflammation [59]. IBD is a typical chronic inflammatory disease of the GI tract; recently, the number of IBD patients has been increasing worldwide. Inflammatory cells, such as lymphocytes and neutrophils, are known to infiltrate the active mucosal lesions of UC patients. These cells are one source of oxidative stress, which induces active mucosal lesions [60]. Single-nucleotide polymorphisms (SNPs) in the antioxidant gene GPX1 are reported to be correlated with UC, while the SOD2 gene was relatively strongly correlated with CD, suggesting that these diseases may be genetically associated with oxidative stress [61]. The mitochondrial DNA mutations in UC and colitic cancer have been analyzed, with marked accumulation noted in UC and even greater accumulation noted in colitic cancer tissue and nontumor adjacent colon tissue [62]. In these same patients, the oxidative stress-related 8-OHdG levels in UC colon tissue were higher than those in control cases, showing that the accumulation of mitochondrial DNA and oxidative stress are strong factors inducing carcinogenesis. In CD, the oxidative DNA damage has been reported to be worse than in controls but the same as in UC [63]. Oxidative stress-related damage to DNA, promoting oncogenic pathways, is therefore sustained to roughly the same degree in UC and CD.

Oxidative stress is an important factor associated with the development of colitis-associated CRC; however, the detailed mechanism of carcinogenesis in IBD remain unclear. Further studies on the relationship between colitis-associated CRC and oxidative stress are needed.

### 5.3. How to Manage Oxidative Stress to Prevent CRC Development: Before Carcinogenesis

Several reports have described the effects of antioxidants, such as vitamin E and selenium, in preventing tumorigenesis. The Linxian trial may have been the first study to investigate the putative prevention of cancer through the use of antioxidants with negative impacts on CRC prevention [38]. However, a meta-analysis reported that preventative effects were observed in sporadic CRC [64]. The authors concluded that a high level of vitamin D was associated with a decreased risk of colorectal adenoma, including recurrent adenoma, which is a pre-CRC lesion in cases of sporadic CRC. Many antioxidant agents have been shown to have anti-colon-carcinogenesis effects in vitro and in vivo.

### 5.4. How to Manage Oxidative Stress in CRC: After Carcinogenesis

Oxidative stress-related molecules have been shown to be involved in colon cancer development, and Nox organizer 1 knockout was shown to reduce tumor-forming ability of colon cancer cells [65]. These facts indicate that antioxidant agents can be used as anti-colon cancer agents. Polyphenols have been reported to inhibit the proliferation of colon cancer cell lines and their migration activity [66].

Given that oxidative stress can lead to the preferential killing of cancer cells, the application of oxidants has been considered as a new approach to CRC therapy. The promotion of oxidative stress overload might be an important strategy to consider in the development of new anticancer drugs. In fact, numerous anticancer drugs used for CRC patients are involved in oxidative stress production [67]. One explanation for the effect of high-dose ascorbic acid on CRC cell lines is that it is dependent on the expression of p53, a tumor suppressor [68]. p53 expressing CRC cell lines were shown to be sensitive to ascorbic acid-induced ROS generation and cell death, while p53-negative cells showed less response. The induction p53 resulted in increased sensitivity to ascorbic acid-induced cell death suggesting the strong correlation between ROS and p53. One another report on the effect of ascorbic acid showed that the CRC cells carrying mutant oncogenes of KRAS or BRAF were killed with high dose ascorbic acid [69]. Dehydroascorbate (DHA), the oxidized form of ascorbic acid, was transported into CRC cells via glucose transporter GLUT1, which was overexpressed in KRAS or BRAF mutant CRC. The increased uptake of DHA caused oxidative stress as intracellular DHA was reduced to ascorbic acid, depleting glutathione. This mechanism might explain why high-dose ascorbic acid can induce cancer cells death, but not normal cell deaths. In these experiments, vitamin E analogue erased the cancer cell toxic response of ascorbic acid via the recovery of the TCA cycle and ATP production [69].

These contradictory reports have important implications for potential anticancer strategies that aim to modulate oxidative stress levels. Thus, further studies are needed to fully elucidate the mechanisms through which antioxidants exert their effects in different types of CRC.

## 6. Oxidative Stress in the Liver

While HBV- and HCV-related chronic hepatitis variants have long been the main background of hepatocarcinogenesis, the incidence of nonalcoholic steatohepatitis (NASH)-related HCC is increasing.

Oxidative stress is one of the features induced by chronic hepatic inflammation in chronic liver disease. As the liver is a mitochondria-rich organ, the damage induces a large amount of ROS; thus, oxidative stress can even be detected in a patient’s serum (Figure 3).

Pre-liver cancer state: Hepatitis B virus (HBV), hepatitis C virus (HCV), and lipid in the liver of nonalcoholic fatty liver disease (NAFLD) induce ROS resulting in chronic inflammation, liver fibrosis, and hepatocarcinogenesis. HBx protein can upregulate oncogenes, such as Ras/Raf/MAPK and PI3K/Akt, and can also disrupt tumor suppressor p53 indicating its oncogenic potential. The HCV core protein could directly interact with the mitochondria resulting in the oxidization of the glutathione pool and reduced NADPH content resulting in increased ROS generation. The fatty acid in NAFLD can induce ROS. Adenosine monophosphate-activated protein kinase (AMPK) which is a heterodimeric serine-threonine kinase that functions as a bridge for energy homeostasis to cellular proliferation, survival and carcinogenesis is one of the key molecules involved in NAFLD progression.

Liver cancer state: A recently established anti hepatocellular carcinoma agent, Sorafenib, is a multikinase inhibitor that can inhibit tumor cell proliferation and angiogenesis through the inactivation of vascular endothelial growth factor receptor (VEGFR), platelet-derived growth factor receptor (PDGFR), c-kit receptor, and the serine-threonine kinase Raf-1. Advanced glycation end products (AGEs) are produced under high sustained glucose conditions and the receptor for AGE (RAGE) has been shown to be overexpressed in HCC. RAGE has been shown to induce sorafenib resistance in in vitro and in vivo xenograft models and the reduction of RAGE was shown to result in an increase in autophagy and a reduction of sorafenib resistance via the activation of the AMPK pathway.

### 6.1. Oxidative Stress in Hepatitis Virus-Related Hepatocarcinogenesis: Before Carcinogenesis

Since HBV and HCV are not cytopathic viruses, immune reactions play a central role in the development of chronic hepatitis [70,71]. The chronic inflammation and liver fibrosis caused by chronic HBV and/or HCV infection contributes to the development of HCC [72]. However, relatively weak oxidative stress has been shown with the analysis of serum samples from patients with HBV-related HCC, while oxidative stress-related markers are significantly increased in patients with HCV-related HCC [73]. One reason for this might be that other features have greater effects in HBV, such as the induction of mutations in oncogenic and tumor suppressor genes by viral proteins, which is frequently observed in noncirrhotic HBV carriers [74,75,76].

Recent progress in antiviral treatments has enabled the control of viral hepatitis, creating a condition of inactive hepatitis. Furthermore, HCV can be eradicated in almost all patients using presently available direct-acting antiviral agents (DAAs). In these patients, hepatocarcinogenesis can be induced by remnant fibrosis as well as environmental factors, such as concomitant fatty liver. Oxidative stress is involved in chronic hepatitis, irrespective of HBV or HCV infection; however, each virus has specific mechanisms that are correlated with the carcinogenic response.

#### 6.1.1. Oxidative Stress in HBV-Related Hepatocarcinogenesis: Before Carcinogenesis

HBV is a DNA virus containing a relaxed circular DNA genome enclosed by an envelope protein [77,78]. After the envelopment and release of mature virions, HBV is converted into a covalently closed circular (ccc) DNA that persists in the nucleus of infected cells that are difficult to eradicate even after clinical cure [79]. The HBV genome encodes several gene products, including the HBx protein and Pre-S protein, which have been shown to be correlated with oxidative stress. 

HBx can upregulate oncogenes such as Ras/Raf/MAPK, PI3K/Akt and can also disrupt tumor suppressor p53 indicating its oncogenic potential [80,81]. A localization analysis of HBx revealed its coexistence with mitochondria [82]. HBx protein binds to the voltage-dependent anion channels (VDAC)-3 in mitochondria and alters the mitochondrial transmembrane potential, resulting in an in ROS and the activation of several transcription factors [83]. The C-terminal region of HBx produced by HBx truncation is required for ROS production [84] and is found in 46% of HCC, but not in nontumor tissues [85]. Although the HBx protein used in the in vitro analysis proved the induction of ROS and hepatocarcinogenesis, the observation of hepatitis B-induced ROS in hepatocarcinogenesis in the clinical setting is rare. There is a report on the in vitro analysis and the analysis of patient samples to investigate the root of HBV-ROS-HCC. HBV-induced mitochondrial ROS mediated DNA methylation resulting in the silencing of SOCS3 and the proliferation of the HCC cell line. SOCS3 was decreased in patients with HBV-associated HCC, indicating the root of HBV-ROS-HCC in these patients [86]. The direct proof of HBV-induced ROS results in hepatocarcinogenesis is difficult.

Reports on HBV genome sequencing have shown several mutations that might be associated with hepatocarcinogenesis [87,88,89,90]. One of the HCC-associated mutations located is in the pre-S region [91]. The pre-S mutant products induce the accumulation of mutated HBV-L protein in the endoplasmic reticulum (ER) of hepatocytes. These mutated proteins have been shown to be correlated with increased ER stress and ROS [91]. The Pre-S mutated proteins in the ER can trigger c-Raf-1/Erk-2 signaling followed by AP-1 activation and the increased expression of hepatic tumors in transgenic models [92].

#### 6.1.2. Oxidative Stress in HCV-Related Hepatocarcinogenesis: Before Carcinogenesis

HCV is an RNA virus coded into structural proteins and nonstructural proteins that have been shown to be oncogenic under some experimental conditions, although these effects might not be strong enough to be clinically evident, as HCV-related HCC is usually found in patients with cirrhosis. HCV antigens, especially core protein, play major roles in pathogenesis and hepatocarcinogenesis of patients with chronic hepatitis C [93]. In HCV protein expressing transgenic mice, the HCV core protein could directly interact with the mitochondria resulting in the oxidization of the glutathione pool and reduced NADPH content [94]. Although weak, the carcinogenic effects of the HCV proteins occur with chronic inflammation and oxidative stress, resulting in hepatocarcinogenesis in elderly patients [95,96].

One of the environmental factors that induce oxidative stress is the iron-related Fenton reaction, which produces a highly toxic forms of ROS: hydroxyl radicals. Because some investigators have reported that phlebotomy and a low-iron diet reduce the risk of HCC in patients with chronic hepatitis C infection, iron toxicity is thought to be involved in hepatocarcinogenesis [97,98]. The oxidative stress induced by HCV reduces hepcidin transcription and subsequently ferroportin expression in enterocytes and increases the uptake of iron in the duodenum [99,100]. Hepatocarcinogenesis was induced in HCV transgenic mice that were fed an excess amount of iron [101]. Iron deposition is closely correlated with the progress of chronic hepatitis C and HCC development; thus, the iron uptake should be controlled. As described above, the lipid-related, direct prooxidant functions of HCV proteins—especially the core and iron-accumulating functions—indicate the importance of the relationship between oxidative stress and hepatocarcinogenesis in patients with chronic hepatitis C.

### 6.2. Oxidative Stress in NAFLD-Related Hepatocarcinogenesis: Before Carcinogenesis

Most patients with NAFLD exhibit nonprogressive NAFL, while a certain proportion of patients progress to NASH, which is broadly defined by the presence of steatosis with inflammation, hepatocyte degeneration and liver fibrosis [102,103], which lead to cirrhosis and HCC [104,105,106]. The pathophysiology of NASH is considered to involve a “two-hit” theory [107]. The first hit is hepatocyte steatosis, which is characterized by the accumulation of triglycerides in the hepatocytes. The second hit consists of various types of cellular stresses, such as apoptosis, oxidative stress, ER stress, and intestinal conditions. The 8-OHdG (an oxidative stress-related marker) expression in the liver tissue is related to the pathological features of NAFLD and hepatocarcinogenesis indicating its importance in disease progression [108]. However, even in the early stages of NAFLD, immune cells have been shown to be evident and several cytokines in the serum and liver, have been shown to be increased [109]. These facts led us to hypothesize that the disease pathogenesis is explained by the “multiple parallel hits theory”. Both the “two-hit theory” and “multiple parallel hits theory” may be included.

Adenosine monophosphate-activated protein kinase (AMPK) which is a heterodimeric serine-threonine kinase that functions as a bridge for energy homeostasis to cellular proliferation, survival and carcinogenesis is one of the key molecules involved in the progression [110]. The synthesis of AMPK is activated by various cellular stressors that deplete ATP, including respiratory chain dysfunction, mitochondrial ATP synthase distress, and oxidative stress [111]. Its activated form (phospho [p]-AMPK) is downregulated in HCC tissues, and low levels of p-AMPK are correlated with a poor prognosis, underscoring the importance of AMPK signaling in HCC [112]. AMPK signaling has recently been reported to be correlated with inflammatory responses, as the energy metabolism in immune cells is related to immunoregulation [113]. Under many oxidative stress-related conditions, including diabetes and dyslipidemia, diminished AMPK activity has been shown to be associated with tissue oxidative stress [114,115]. 

As mentioned above, oxidative stress is linked to the progression from NAFLD to HCC. The correlation of AMPK and clinical conditions encourages us to control AMPK as a cancer regulator.

Iron depletion improves not only iron overload but also insulin resistance, suggesting that iron toxicity is involved in several metabolic pathways [116].

The guidelines for NAFLD recommend the use of antioxidant therapies, such as vitamin E, for NASH, however, whether this treatment reduces the risk for hepatocarcinogenesis is not clear [117,118]. Many other antioxidant agents may show favorable effects on NASH and NASH-related hepatocarcinogenesis. 

Antidiabetic agents are recommended for patients whose condition is complicated by diabetes. Of the antidiabetic agents, metformin, pioglitazone, glucagon-like peptide 1 receptor agonists (GLP-1 RAs), and dipeptidyl-peptidase-4 inhibitors (DPP-4Is) have been shown to have favorable effects on NAFLD [119,120]. Metformin and pioglitazone are regarded as antioxidant agents. Metformin has also been shown to activate AMPK by inhibiting mitochondrial complex I; this function is considered to be important for its action [121]. Metformin-related AMPK pathway activation is involved in many cell types including T cells, B cells, hepatocytes, and even liver fibrosis-inducing hepatic stellate cells (HSCs). In in vitro and in vivo models (mice), metformin suppressed the expression of alpha smooth muscle actin (α-SMA) via the activation of AMPK activation and the inhibition of the succinate-related pathway in HSCs [122]. HSC activation is involved also in hepatocarcinogenesis. In addition, metformin activates the nuclear factor erythroid 2-related factor (Nrf2) signaling pathway, resulting in the production of heme oxygenase-1 (HO-1; an antioxidant enzyme), in human endothelial cells [123]. Metformin is one of the most important agents and its long-term effects should be investigated to determine whether it is associated with a reduced risk of HCC development.

Pioglitazone is a thiazolidinedione that activates peroxisome proliferator-activated receptor (PPAR) γ, improving insulin resistance. A six-month randomized study of pioglitazone revealed a histopathological reduction in liver necroinflammation with no reduction in fibrosis [124].

The administration of L-carnitine (a mitochondrial long chain fatty acid uptake related molecule) was reported to be associated with the histological improvement of NASH in a randomized controlled study [125].

Flavonoids (heterogeneous polyphenols) have been shown to have an antioxidant function and to protect the liver from ROS-induced damage [126]. A mixture of flavonolignans and minor polyphenolic compounds derived from the milk thistle plant (*Silybum marianum*) named silymarin used as an antioxidative agent [127]. The main component of silymarin, silybin, has been shown to restore nicotinamide adenine dinucleotide (NAD+) levels, resulting in the improvement of NAFLD [128]. Silymarin was shown to be effective for improving NASH-related fibrosis in a randomized, double-blind, placebo-controlled study [129].

### 6.3. How to Manage Oxidative Stress in HCC: After Hepatocarcinogenesis

After the development of HCC, the role of oxidative stress changes. An adequate degree of oxidative stress should be maintained in order to regulate the progression of HCC. The ideal choice of agent for controlling chronic hepatitis or related hepatocarcinogenesis without suppressing the physiological roles of oxidative stress is difficult to determine.

Sorafenib has been widely accepted as an effective oral chemotherapeutic agent for advanced HCC. Sorafenib is a multikinase inhibitor that can inhibit tumor cell proliferation and angiogenesis through the inactivation of vascular endothelial growth factor receptor (VEGFR), platelet-derived growth factor receptor (PDGFR), c-kit receptor, and the serine-threonine kinase Raf-1 [130]. Recently, other multi-tyrosine-kinase inhibitors (TKI), including lenvatinib, regorafenib, cabozantinib, and ramucirumab, have also been developed for the treatment of HCC [90,91,92]. Given that recent clinical experience suggests that sorafenib resistance may develop after long-term administration, it is important to investigate the possibility of inducing sorafenib resistance with an antioxidant or prooxidant approach. Autophagy, phosphoinositide 3-kinase (PI3K)/Akt signaling and the advanced glycation end products (AGE) pathway are reported to be involved with the acquisition of resistance to sorafenib in HCC [131]. Given that these pathways are associated with AMPK/mTOR signaling, metformin is a candidate agent for reducing sorafenib resistance. AGEs are produced under high sustained glucose conditions and the receptor for AGE (RAGE) has been shown to be overexpressed in HCC. RAGE has been shown to induce sorafenib resistance in in vitro and in vivo xenograft models and the reduction of RAGE was shown to result in an increase in autophagy and a reduction of sorafenib resistance via the activation of the AMPK pathway. Although metformin is expected to be effective for reducing sorafenib resistance, one clinical study reported a bad outcome in patients treated with metformin and sorafenib [132].

Oxidative stress inducing agents are a next candidate for improving the effects of sorafenib. Tetrandrine, a bisbenzylisoquinoline alkaloid that increases ROS, has been shown to exert synergistic antitumor activity with sorafenib [133]. More data on combination therapies with sorafenib will be required to draw more solid conclusions.

## 7. Oxidative Stress in Pancreaticobiliary Tract Diseases

Pancreaticobiliary malignancies exhibit poor prognoses [49]. Recently, new chemotherapies have been developed; however, they are not expected to cure these diseases. Pancreaticobiliary malignancies are related to obesity as well as other cancers [134,135]. Pancreaticobiliary malignancies have also been reported to be correlated with oxidative stress conditions [136,137]. We previously reported that the oxidative balance in patients with pancreaticobiliary malignancies was related to their prognoses [138]. However, as pancreaticobiliary cancer high risk patients are not followed with care, the precise relationship remains poorly understood.

### 7.1. Oxidative Stress in Pancreatic Carcinogenesis: Before Carcinogenesis

The maintenance of homeostasis has recently been accepted as important for pancreatic cancer (PC) patients, as these patients often develop a cachexic status after pancreaticoduodenectomy surgery or toxic chemotherapies. Sufficient muscle mass has recently been indicated as an important factor that predicts good survival after pancreaticoduodenectomy [139]. The muscle quality, such as the fat-infiltrated muscle, known as myosteatosis, has also been indicated to be a survival-defining factor after pancreaticoduodenectomy [140]. A transcriptomic analysis revealed that oxidative phosphorylation was disrupted in myosteatotic muscles [141]. Oxidative stress is a condition that influences the environment of cancer cells, including PC cells [142].

### 7.2. Oxidative Stress in Biliary Tract Carcinogenesis: Before Carcinogenesis

Oxidative stress is strongly related to biliary tract carcinogenesis as well as PC. The risk factors for biliary carcinoma include parasitic infections, primary sclerosing cholangitis, biliary duct cysts, hepatolithiasis and toxin exposure. Other less-established potential risk factors include inflammatory bowel disease, HCV and HBV disease, liver cirrhosis, diabetes, obesity, alcohol consumption, and smoking [143]. Although direct biliary tract injury is an accepted risk factor, the potential risk factors are correlated with chronic inflammation and oxidative stress. In a previous study, we reported that an oxidative imbalance influences the prognosis of patients with biliary carcinoma and revealed the correlation between oxidative stress and carcinogenesis using a diabetes-based hepatocarcinogenesis model (the STAM mouse) that was co-administered the cholangiotoxic agent alpha naphthyl isothiocyanate (ANIT) [138]. In that study, we also reported that the intestinal microbiota was correlated with oxidative stress via bile acid changes (an increase in the secondary-to-primary bile acid balance was observed). The secondary bile acid deoxycholic acid (DCA) is known to cause DNA damage and is increased by dietary or genetic obesity; this indicates that increased secondary bile acids induce oxidative stress and create an environment conducive to cancer [144,145].

Epidemiological studies have recently shown that metformin use decreased incidence of pancreatic cancer [146]. Although the preferable effect depends partly on its body weight reduction, it showed significant reduction in cancer incidence [147].

### 7.3. How to Manage Oxidative Stress in Pancreaticobiliary Tract Cancers: After Carcinogenesis

Oxidative stress is a crucial factor in the cancer microenvironment, but oxidative stress management in pancreaticobiliary tract cancers remains controversial. The ability of several antioxidant agents to control oxidative stress for cancer prevention or treatment is being investigated in vitro and in vivo. Vitamin E is an antioxidant agent and a potential chemopreventive agent in cancer cells [148,149]. Husain et al. reported on the efficacy of vitamin E as a therapeutic agent in a mouse model of PC [150]. These reports indicate that antioxidative therapy has potential applications in the treatment of PC.

However, in order to administer antioxidants to cancer patients, we must carefully consider the oxidative stress-related environment. Several studies have indicated unfavorable outcomes of antioxidant vitamin A or E therapy [119]. Thus, the concept of controlling oxidative stress in this manner must be reevaluated. It is difficult to control carcinogenesis without suppressing the physiological role of oxidative stress [151]. These controversial opinions may indicate that oxidative stress should be moderately controlled, as opposed to drastically regulated [149].

Recently, mitochondria-targeted analogs of metformin have been shown to enhance antiproliferative and radiosensitizing effects in pancreatic cancer cells [152]. One of the set of mitochondria-targeted metformin analogs synthesized by attaching alkyl chain containing a triphenylphosphonium cation to metformin was found to be 1000 times more efficacious than metformin in inhibiting pancreatic cancer cell lines. This agent could inhibit mitochondrial complex I and stimulate superoxide and AMPK resulting in cell death. Its effects in normal cell lines were minimal. This is expected to be a future approach in pancreatic cancer treatment.

The antioxidant molecule HO-1 was increased in pancreatic cancer tissue and the downregulation of HO-1 resulted in sensitization to gemcitabine treatment, indicating that oxidative stress is important for controlling pancreatic cancer [153]. Oxidative stress-inducing agents may also be able to repair intrinsic resistance to chemotherapy, such as resistance to gemcitabine, in PC patients [154].

## 8. Conclusions

Oxidative stress is a disease-advancing reaction; however, as it also has an efficient anticancer function, this balance must be regulated with great care. The oxidative–antioxidative stress condition differs among cancers and chronic inflammatory diseases carrying a cancer risk. Determining the oxidative stress load via serum markers or histological analyses is an important method for assessing a patient’s condition. By evaluating a patient’s oxidative stress load, we can decide how to manage their condition-via either oxidative stress eliminators or antioxidants or through an alternative approach, such as the administration of high-dose ascorbic acid. However, there is no definite clinical data showing the favorable effects of antioxidant administration and there is no direct proof of the role of oxidants in tumor growth. We must therefore conclude that there is no solid proof of the causative role of ROS in carcinogenesis or tumor growth in human subjects.

## Figures and Tables

**Figure 1 cancers-11-00213-f001:**
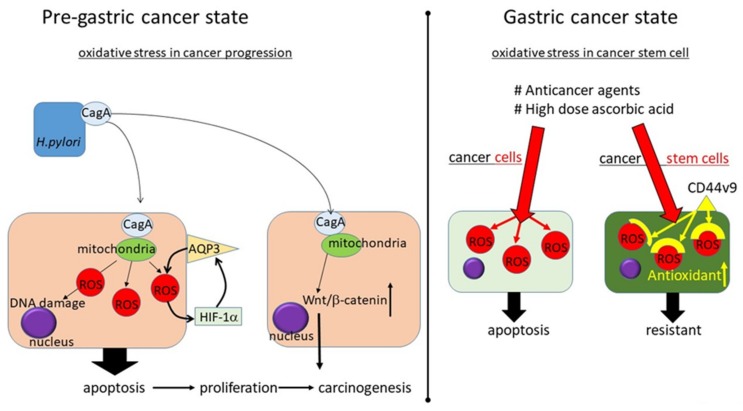
A conceptual diagram of the oxidative stress in the upper gastrointestinal tract.

**Figure 2 cancers-11-00213-f002:**
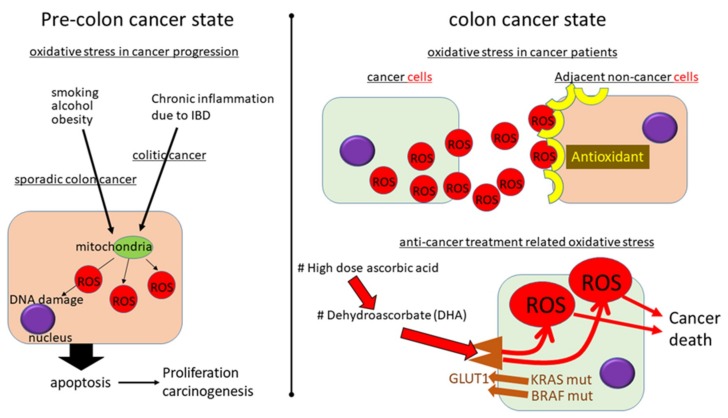
A conceptual diagram of the oxidative stress in colorectal cancer (CRC).

**Figure 3 cancers-11-00213-f003:**
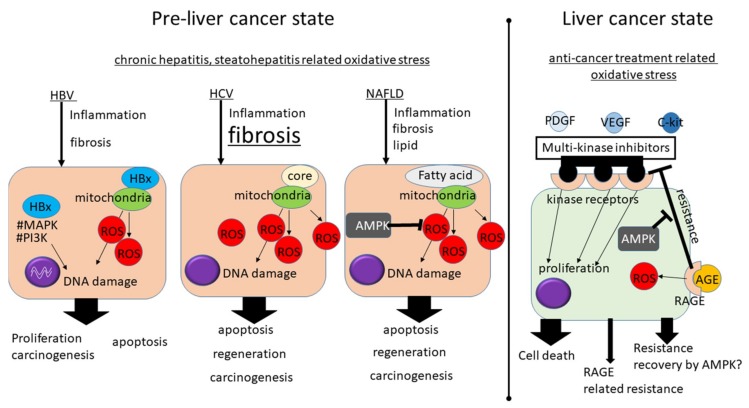
A conceptual diagram of oxidative stress in liver cancer.

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
