# Peer review of "Paradoxical Roles of Oxidative Stress Response in the Digestive System before and after Carcinogenesis"

_cancers, 2019, doi:10.3390/cancers11020213_

Round 1
Reviewer 1 Report
In the submitted manuscript the authors review the role of oxidative stress in carcinogenesis in the digestive system. The topic should be of interest to the cancer researchers, as the authors cite many clinical trials with vitamin-based antioxidants.
The authors should consider the following points for the revision of the manuscript:
1. While mitochondria are often regarded as a major cellular source of ROS, there are hundreds of papers implicating NADPH oxidases (Nox) proteins as a source of ROS, used by cancer cells to drive proliferation. NADPH oxidases are the only enzymes, whose only known function is to produce ROS. For example, numerous reports implicate Nox1 in the development and progression of colon cancer.
2. The authors should add a paragraph discussing the evidence for the possible sources of ROS.
3. It would be beneficial, if the authors discuss the possible targets of ROS, which could drive cancer cell proliferation and cancer cell death during cancer therapy.
4. A section on experimental approaches and limitations to monitor oxidative stress in the in vivo models and in patients would be beneficial.
5. The ability of antioxidants to directly scavenge ROS in vivo is questionable, as the PK properties of many antioxidants are unfavorable, and it is more than likely that polyphenolic compounds work via modulation of cellular signaling, by modulation of the activity of kinases and phosphatases, rather than by scavenging ROS. The strategy to inhibit ROS formation is, therefore, more promising than ROS scavenging. Also, activation of enzymatic antioxidant system (via Nrf2 pathway) may provide a way to decrease oxidant stress. These possibilities should be discussed.
6. Based on the cited clinical trials using antioxidants, one can conclude that there is no solid proof for the causative role of ROS in carcinogenesis or tumor growth in human subjects. This should be discussed in the conclusions section. As stated by the authors (lines 148-150): ‘It is still difficult to draw conclusions about the effectiveness of antioxidant treatment…’
7. Also, the experimental preclinical in vivo models do not provide a direct proof for the role of oxidants in tumor growth. Most data show the correlation of tumor growth with ROS, as a product of inflammatory states, but there is no direct proof that selective modulation of ROS would inhibit tumor growth. Inflammation is associated not only with increased oxidant production, but also with increased levels of various cytokines and chemokines. Some of them are known to promote tumor growth and metastasis.
8. Line 46 – “production of oxidative stress” – please rewrite (‘production of ROS’, or just ‘oxidative stress’
9. Figure 1 – please make it more clear – It is difficult to understand the figure.
10. Table 1 – each piece of information should be accompanied by a reference to the literature.
11. Many claims in the text are not accompanied by the original source – please include references to the claims regarding the role of ROS in carcinogenesis. For example, sentences on: lines 105; 203; 223; 298;
12. Lines 115-116 – please clarify how increased apoptosis induces cell proliferation and development of cancer stem cells.
13. Line 134 – as discussed later, ascorbic acid should be regarded as possible pro-oxidant rather than antioxidant, due to its susceptibility to undergo autooxidation and production of hydrogen peroxide. Thus, the experimental setup may have been not optimal for the determination of the role of antioxidants in gastric tumorigenesis.
14. Lines 167-177 – it is not clear why ascorbic acid would need to be administered iv, if the target is gastric cancer, one would expect a better result with oral administration.
15. Line 253 – vitamin D does not work as an antioxidant.
16. Line 304 – please specify the markers used.
17. Section 4.1.1 – are there any data implicating ROS in HBV-related hepatocarcinogenesis?
18. It is not clear why the authors discuss the effects of curcumin in the review. Are there any data suggesting that the effects observed are due to antioxidant action of this compound?
19. Iron is not only a mediator of the Fenton reaction, but is important to the cell bioenergetics and synthesis of biomolecules and thus for cell proliferation. In fact, some experimental anticancer drugs (e.g. triapine) are based on their ability to chelate iron and inhibit ribonucleotide reductase.
20. The role of oxidants in AMPK activation/inhibition is not clear. There are reports suggesting that mitochondrial oxidants can activate AMPK. Also, metformin is assumed to activate AMPK by inhibiting mitochondrial complex I, rather than by modulation of cellular oxidants.
21. Recent reports suggest that induction of mitochondrial ROS production using mitochondria-targeted agents (e.g., mito-metformin, mito-honokiol, mito-vitamin E) can bloack proliferation of cancer cells in vitro and tumor growth in vivo (e.g. inhibition of pancreatic cancer by mitochondria-targeted metformin). The authors may consider adding a paragraph on a perspective use of targeted pro-oxidants for the treatment of the tumors of digestive system.
Author Response
To the Editors and Reviewer1 of Cancers:
Re: cancers-430085
We are grateful to the reviewer for the critical comments and useful suggestions that have helped us improve our paper. As indicated in the responses that follow, we have taken all these comments and suggestions into account in the revised version of paper.
We hope that the revised version will now be deemed suitable for publication in Cancers.
Sincerely,
Akinobu Takaki
2-5-1 Shikata-cho, Okayama City, Okayama 700-8558, Japan
Tel: +81-86-235-7220; Fax: +81-86-225-5991
E-mail: akitaka@md.okayama-u.ac.jp
Reviewer 1
In the submitted manuscript the authors review the role of oxidative stress in carcinogenesis in the digestive system. The topic should be of interest to the cancer researchers, as the authors cite many clinical trials with vitamin-based antioxidants.
The authors should consider the following points for the revision of the manuscript:
While mitochondria are often regarded as a major cellular source of ROS, there are hundreds of papers implicating NADPH oxidases (Nox) proteins as a source of ROS, used by cancer cells to drive proliferation. NADPH oxidases are the only enzymes, whose only known function is to produce ROS. For example, numerous reports implicate Nox1 in the development and progression of colon cancer.
Response: We added several explanations for Nox. (Line 40-43, 207-210, 321-323).
2. The authors should add a paragraph discussing the evidence for the possible sources of ROS.
Response: We added one additional paragraph explaining the sources of ROS. (Line 35-43).
3. It would be beneficial, if the authors discuss the possible targets of ROS, which could drive cancer cell
proliferation and cancer cell death during cancer therapy.
Response: We added one additional paragraph explaining the targets of ROS. (Line 49-53).
4. A section on experimental approaches and limitations to monitor oxidative stress in the in vivo models
and in patients would be beneficial.
Response: We added one additional paragraph explaining the in vivo and patient monitoring markers in a new section 3.2. Problems in the management of oxidative stress. (Line 104-112).
5. The ability of antioxidants to directly scavenge ROS in vivo is questionable, as the PK properties of many
antioxidants are unfavorable, and it is more than likely that polyphenolic compounds work via modulation
of cellular signaling, by modulation of the activity of kinases and phosphatases, rather than by scavenging
ROS. The strategy to inhibit ROS formation is, therefore, more promising than ROS scavenging. Also,
activation of enzymatic antioxidant system (via Nrf2 pathway) may provide a way to decrease oxidant
stress. These possibilities should be discussed.
Response: We added one additional paragraph explaining the above mentioned opinion in a new section 3.2. Problems in the management of oxidative stress. (Line 113-120).
6. Based on the cited clinical trials using antioxidants, one can conclude that there is no solid proof for the
causative role of ROS in carcinogenesis or tumor growth in human subjects. This should be discussed in
the conclusions section. As stated by the authors (lines 148-150): ‘It is still difficult to draw conclusions
about the effectiveness of antioxidant treatment…’
Response: We added one additional paragraph explaining the above-mentioned opinion in the conclusion. (Line 592-595).
7. Also, the experimental preclinical in vivo models do not provide a direct proof for the role of oxidants in
tumor growth. Most data show the correlation of tumor growth with ROS, as a product of inflammatory
states, but there is no direct proof that selective modulation of ROS would inhibit tumor growth.
Inflammation is associated not only with increased oxidant production, but also with increased levels of
various cytokines and chemokines. Some of them are known to promote tumor growth and metastasis.
Response: We added one additional paragraph explaining the above-mentioned opinion in the conclusion. (Line 592-595).
8. Line 46 – “production of oxidative stress” – please rewrite (‘production of ROS’, or just ‘oxidative stress’
Response: We changed this as indicated.
9. Figure 1 – please make it more clear – It is difficult to understand the figure.
Response: We changed the figure to explain each section-upper GI, colon, and liver. Given that the data on pancreatico-biliary diseases are short, we didn’t make a figure for this section.
10. Table 1 – each piece of information should be accompanied by a reference to the literature.
Response: Given that we changed Figure 1 to Figures 1 to 3 to show each explanation for each organ section, we deleted Table 1.
11. Many claims in the text are not accompanied by the original source – please include references to the claims regarding the role of ROS in carcinogenesis. For example, sentences on: lines 105; 203; 223; 298;
Response: In accordance with the reviewer’s suggestion, we added several articles.
12. Lines 115-116 – please clarify how increased apoptosis induces cell proliferation and development of cancer stem cells.
Response: We changed the sentence as follows and added one additional reference.
The oxidative stress produced by H. pylori infection, especially with CagA, not only causes DNA damage but also prevents DNA repair mechanisms from functioning properly, subsequently causing increased apoptosis and cellular proliferation. In addition, CagA upregulates Wnt/b-catenin signaling resulting in the development of cancer stem cells {Yong, 2016 #216}. (Line 171-175).
13. Line 134 – as discussed later, ascorbic acid should be regarded as possible pro-oxidant rather than antioxidant, due to its susceptibility to undergo autooxidation and production of hydrogen peroxide. Thus, the experimental setup may have been not optimal for the determination of the role of antioxidants in gastric tumorigenesis.
Response: Based on this suggestion and the increase in the contents, we deleted the paragraph.
14. Lines 167-177 – it is not clear why ascorbic acid would need to be administered iv, if the target is gastric cancer, one would expect a better result with oral administration.
Response: We added following explanation. “a pharmacologic dose of ascorbic acid which could not be achieved via oral intake,”(Line 234).
15. Line 253 – vitamin D does not work as an antioxidant.
Response: We deleted the passage on vitamin D and C.
16. Line 304 – please specify the markers used.
Response: We deleted the wrong marker.
17. Section 4.1.1 – are there any data implicating ROS in HBV-related hepatocarcinogenesis?
Response: We added an explanation about the direct proof of ROS in HBV-related hepatocarcinogenesis. (Line 404-408)
18. It is not clear why the authors discuss the effects of curcumin in the review. Are there any data suggesting that the effects observed are due to antioxidant action of this compound?
Response: As you indicated, the effect of curcumin is complex. We deleted the explanation about curcumin.
19. Iron is not only a mediator of the Fenton reaction, but is important to the cell bioenergetics and synthesis of biomolecules and thus for cell proliferation. In fact, some experimental anticancer drugs (e.g. triapine) are based on their ability to chelate iron and inhibit ribonucleotide reductase.
Response: There was insufficient space to discuss iron more precisely because of the word limit, we therefore concentrated on the involvement of iron in the Fenton reaction.
20. The role of oxidants in AMPK activation/inhibition is not clear. There are reports suggesting that mitochondrial oxidants can activate AMPK. Also, metformin is assumed to activate AMPK by inhibiting mitochondrial complex I, rather than by modulation of cellular oxidants.
Response: In accordance with the reviewer’s suggestion, we added following explanation and references.
The synthesis of AMPK is activated by various cellular stressors that deplete ATP, including respiratory chain dysfunction, mitochondrial ATP synthase distress, and oxidative stress {Kahn, 2005 #219}.(Line 453-455).
Metformin has also been shown to activate AMPK by inhibiting mitochondrial complex I; this function is considered to be important for its action {Owen, 2000 #218}. (Line 473-475).
21. Recent reports suggest that induction of mitochondrial ROS production using mitochondria-targeted agents (e.g., mito-metformin, mito-honokiol, mito-vitamin E) can bloack proliferation of cancer cells in vitro and tumor growth in vivo (e.g. inhibition of pancreatic cancer by mitochondria-targeted metformin). The authors may consider adding a paragraph on a perspective use of targeted pro-oxidants for the treatment of the tumors of digestive system.
Response: We added an explanation about mito-metformin in the pancreatic cancer section.
(Line 573-579).
Reviewer 2 Report
The authors in the present review highlight the paradoxical effects of oxidative stress and antioxidant agents in the digestive system before and after carcinogenesis.
The review is well written in the present form and can be accepted after minor revision.
Minor revision:
Figure 1 can be improved for a better understanding.
The authors should include in the discussions the role of heme oxygenase- 1 (HO-1) in the cancer (Eur J Med Chem. 2017, 142:163-178; J Biomed Sci. 2015, 22:22). References related to the role of HO-1 inducers/inhibitors in the digestive system before and after carcinogenesis may be included (Langenbecks Arch Surg (2016) 401:99–111; Mol Med 23: 215-224, 2017; Molecules 2010,15, 3338-3355)
Author Response
To the Editors and Reviewer2 of Cancers:
Re: cancers-430085
We are grateful to the reviewer for the critical comments and useful suggestions that have helped us improve our paper. As indicated in the responses that follow, we have taken all these comments and suggestions into account in the revised version of paper.
We hope that the revised version will now be deemed suitable for publication in Cancers.
Sincerely,
Akinobu Takaki
2-5-1 Shikata-cho, Okayama City, Okayama 700-8558, Japan
Tel: +81-86-235-7220; Fax: +81-86-225-5991
E-mail: akitaka@md.okayama-u.ac.jp
Reviewer 2
The authors in the present review highlight the paradoxical effects of oxidative stress and antioxidant agents in the digestive system before and after carcinogenesis.
The review is well written in the present form and can be accepted after minor revision.
Minor revision:
1. Figure 1 can be improved for a better understanding.
Response: Given that all three reviewers suggested to improve figure, we changed the figure to explain the mechanisms in each section.
2. The authors should include in the discussions the role of heme oxygenase- 1 (HO-1) in the cancer (Eur J
Med Chem. 2017, 142:163-178; J Biomed Sci. 2015, 22:22). References related to the role of HO-1
inducers/inhibitors in the digestive system before and after carcinogenesis may be included (Langenbecks Arch Surg (2016) 401:99–111; Mol Med 23: 215-224, 2017; Molecules 2010,15, 3338-3355
Response: In accordance with the reviewer’s suggestion, we added the following explanation:
The function of antioxidants is also involved in carcinogensis. Heme oxygenase-1 (HO-1), an anti-oxidant molecule, is an enzyme that catalyzes the oxidative degradation of cellular heme into free iron, carbon monoxide, and biliverdin, which is then converted into non-toxic bilirubin. Targeting HO-1 to reduce the antioxidant function of cancer cells is a recent approach in cancer treatment (Salerno, Eur J Med Chem. 2017). (Line 57-61).
The anti-oxidant molecule HO-1 was increased in pancreatic cancer tissue and the downregulation of HO-1 resulted in sensitization to gemcitabine treatment, indicating that oxidative stress is important for controlling pancreatic cancer. (Line 580-583).
Reviewer 3 Report
The review article “Paradoxical roles of oxidative stress response in the digestive system before and after carcinogenesis” by Takaki et al hardly provides any newer information to the scientific community. In this article, the authors tried to focus on the anomalous role of oxidative stress and antioxidant agent during progression as well as therapeutic intervention of cancer. There are ample amount of review articles available and are focusing particularly on this subject area. Therefore, authors need to highlights and mention the importance of the present study and why their study is unique in comparison to other studies.
Major Comment:
1. Overall, the manuscript is very poorly written, very difficult to follow and needs to rewrite.
2. Figure 1 is not well illustrated and is very hard to follow.
3. Authors need to incorporate more illustration covering each section of the manuscript for reader’s aid.
4. Authors should include a “perspective of review” section in their review article.
5. Authors need to incorporate references in the table 1.
The information packed in this review article are not qualitatively sufficient to be published in this present format.
Author Response
To the Editors and Reviewer3 of Cancers:
Re: cancers-430085
We are grateful to the reviewer for the critical comments and useful suggestions that have helped us improve our paper. As indicated in the responses that follow, we have taken all these comments and suggestions into account in the revised version of paper.
We hope that the revised version will now be deemed suitable for publication in Cancers.
Sincerely,
Akinobu Takaki
2-5-1 Shikata-cho, Okayama City, Okayama 700-8558, Japan
Tel: +81-86-235-7220; Fax: +81-86-225-5991
E-mail: akitaka@md.okayama-u.ac.jp
Reviewer 3
The review article “Paradoxical roles of oxidative stress response in the digestive system before and after carcinogenesis” by Takaki et al hardly provides any newer information to the scientific community. In this article, the authors tried to focus on the anomalous role of oxidative stress and antioxidant agent during progression as well as therapeutic intervention of cancer. There are ample amount of review articles available and are focusing particularly on this subject area. Therefore, authors need to highlights and mention the importance of the present study and why their study is unique in comparison to other studies.
Major Comment:
Overall, the manuscript is very poorly written, very difficult to follow and needs to rewrite.
Response: This manuscript has been checked by a professional editor who is a native speaker of English.
2. Figure 1 is not well illustrated and is very hard to follow.
Response: Given that all three reviewers suggested to improve figure, we changed the figure to explain the mechanisms in each section.
3. Authors need to incorporate more illustration covering each section of the manuscript for reader’s aid.
Response: We changed the figure to explain the mechanisms in each section as reviewer 3 suggested.
4.Authors should include a “perspective of review” section in their review article.
Response: In accordance with the reviewer’s suggestion, we added the “Perspective of this review” section. (Line 63-70)
5. Authors need to incorporate references in the table 1.
Response; Given that we added explanatory figures in each section, we deleted the table.
The information packed in this review article are not qualitatively sufficient to be published in this present format.
Round 2
Reviewer 3 Report
The manuscript entitled “Paradoxical roles of oxidative stress response in the digestive system before and after carcinogenesis” by Takaki et al. have been revised and addressed all the comments. Therefore, this revised manuscript could be accepted by the journal in this present form.